# Predicting major adverse cardiovascular events for secondary prevention: protocol for a systematic review and meta-analysis of risk prediction models

Ralph K Akyea [1], Jo Leonardi-Bee,[2] Folkert W Asselbergs,[3,4] Riyaz S Patel,[4] Paul Durrington,[5] Anthony S Wierzbicki,[6] Oluwaseun H Ibiwoye,[2] Joe Kai,[1] Nadeem Qureshi [1], Stephen F Weng[1]

For numbered affiliations see end of article.

**Correspondence to**
Dr Ralph K Akyea;
mszrka@nottingham.ac.uk

## ABSTRACT

**Introduction** Cardiovascular disease (CVD) is the leading cause of morbidity and mortality globally. With advances in early diagnosis and treatment of CVD and increasing life expectancy, more people are surviving initial CVD events. However, models for stratifying disease severity risk in patients with established CVD for effective secondary prevention strategies are inadequate. Multivariable prognostic models to stratify CVD risk may allow personalised treatment interventions. This review aims to systematically review the existing multivariable prognostic models for the recurrence of CVD or major adverse cardiovascular events in adults with established CVD diagnosis.

**Methods and analysis** Bibliographic databases (Ovid MEDLINE, EMBASE, PsycINFO and Web of Science) will be searched, from database inception to April 2020, using terms relating to the clinical area and prognosis. A hand search of the reference lists of included studies will also be done to identify additional published studies. No restrictions on language of publications will be applied. Eligible studies present multivariable models (derived or validated) of adults (aged 16 years and over) with an established diagnosis of CVD, reporting at least one of the components of the primary outcome of major adverse cardiovascular events (defined as either coronary heart disease, stroke, peripheral artery disease, heart failure or CVD-related mortality). Reviewing will be done by two reviewers independently using the pre-defined criteria. Data will be extracted for included full-text articles. Risk of bias will be assessed using the Prediction model study Risk Of Bias ASsessment Tool (PROBAST). Prognostic models will be summarised narratively. If a model is tested in multiple validation studies, the predictive performance will be summarised using a random-effects meta-analysis model to account for any between-study heterogeneity.

**Ethics and dissemination** Ethics approval is not required. The results of this study will be submitted to relevant conferences for presentation and a peer-reviewed journal for publication.

**PROSPERO registration number** CRD42019149111.

## INTRODUCTION

Cardiovascular disease (CVD), the leading cause of morbidity and mortality, is a

## Strengths and limitations of this study

► This comprehensive systematic review will evaluate the existing literature on prognostic models that have been developed to assess cardiovascular disease (CVD) severity in adults with established CVD diagnosis.
► The constituent predictor variables of prognostic models will be identified and their effectiveness evaluated and reported.
► A potential limitation of this review may be the high level of heterogeneity in available studies.
► Evidence from observational cohort studies may be used in this context and this level of evidence may, therefore, be subject to bias and confounding.
► The difficulty of aggregating quantitative measures from prognostic models with variations in clinical outcome definitions.

significant and ever-growing problem in every region of the world.[1] With advances in diagnosis and treatment of CVD and increasing life expectancy, more people are surviving initial CVD events. For patients with established CVD, the priority is to prevent a subsequent CVD event or premature death. Current secondary prevention interventions have achieved substantial success in reducing the risk of cardiovascular events and mortality after incident CVD events.[2] However, the prognosis of patients with established CVD remains sub-optimal.[3]

Intensified pharmacological therapy, of anti-thrombotic and lipid-lowering medications, is efficacious in these individuals with high residual CVD risk but this could have harmful excess risk in those with low risk. Also, these intensive therapies are expensive, hence the need to be targeted. It is, therefore, important to identify prognostic factors (demographic, clinical and laboratory

characteristics of patients) associated with an increased risk of CVD recurrence or occurrence of a major adverse cardiovascular event (MACE). MACE, an endpoint frequently used in cardiovascular research, remains the major cause of morbidity and mortality in patients living with CVD,[4] hence the most relevant outcome in secondary prevention. MACE is frequently defined as a composite of non-fatal stroke, non-fatal myocardial infarction or cardiovascular death,[5 6] and occasionally expanded to include heart failure, coronary revascularisation and ischaemic cardiovascular events.[7]

Prognostic factors when combined in a prognostic model are generally useful in identifying groups of patients at highest risk of disease occurrence/recurrence (CVD recurrence or MACE outcomes) and, thus, inform preventive interventions, patient counselling, clinical guidelines and policies.[8] Though there has been a significant focus on prognostic models aimed at primary prevention in the general population,[9 10] there has been less progress in developing prognostic models for stratifying CVD severity in patients who already have had an initial CVD event.

We aim to systematically review all the evidence for current prognostic models for stratifying CVD severity based on CVD recurrence or occurrence of a major adverse cardiovascular event in individuals with an established CVD diagnosis. The findings of this review could inform clinical practice and patient care by identifying patient characteristics of consistent prognostic value when adjusted for other prognostic factors and by summarising the current prognostic models and their predictive performance.

## Research aims
This review aims to identify and summarise studies of any design evaluating prognostic models (and clinical decision rules based on such models) that use multiple prognostic factors in combination to predict CVD recurrence or occurrence of major adverse cardiovascular events in patients with an established CVD for secondary prevention.

## METHODS
This systematic review and meta-analysis is being conducted using the methodology recommended for systematic review and meta-analysis of prediction models[11] and Critical Appraisal and Data Extraction for Systematic Reviews of Prediction Modelling Studies: the CHARMS checklist.[12] This review will be reported according to the Preferred Reporting Items for Systematic Reviews and Meta-Analyses checklist.[13] The review is registered in the International Prospective Register of Systematic Reviews (PROSPERO): CRD42019149111 and all subsequent updates to the review will be registered here.

## Selection criteria
### Study design
This review will include multivariable prognostic prediction studies that meet the following criteria:
1. Published as an original research article (that developed, compared or validated a multivariable prognostic model or clinical prediction rule) in a peer-reviewed journal.
2. Used comparative study designs including clinical trials, cohort, case-control and cross-sectional studies.

Studies will be excluded if they were published as conference proceedings, dissertations, case-reports, case-series, reviews, editorials, expert opinions or consensus paper abstracts only.

### Patient group
Adults, 16 years and above, with an established diagnosis of CVD (where CVD is defined as a documented clinical diagnosis of arterial occlusive events, including coronary artery disease, cerebrovascular artery disease and peripheral artery disease (PAD)).[14 15]

### Setting
Studies in any setting will be included.

### Potential prognostic models
Studies must report a prognostic model (derived, validated, or both) using multiple prognostic risk factors in combination to predict CVD recurrence or occurrence of major adverse cardiovascular events in adults with an established CVD diagnosis.

### Primary and secondary outcomes
Major adverse cardiovascular event is the primary outcome and is defined as a record/diagnosis of either coronary artery disease (including myocardial infarction, coronary artery bypass grafting and percutaneous coronary intervention); stroke (including carotid endarterectomy); peripheral arterial disease (including PAD-related complications such as gangrene, amputation); heart failure or CVD-related mortality. The included studies for this review should report results for at least one of the components of the MACE primary outcome.

Secondary outcomes of interest for this review include all-cause mortality, adverse effects related to the management of CVD, health-related quality of life and CVD-related medical encounters (contact with primary care, hospitalisation and referral activities).

### Search strategies
The following databases will be searched: Ovid MEDLINE (R) (1946–present), EMBASE (1883–present), PsycINFO (1860–present) and Web of Science (1998–present) for articles published in peer-reviewed journals. The search terms are presented in online supplementary file—appendix 1 and covers expressions for CVD, risk scores and predictive performance assessment. Hand searches of the reference lists and citation tracking for all relevant identified papers will be carried out for additional

studies that fulfil the aforementioned inclusion criteria. No language restrictions will be applied, and translations will be sought where necessary.

## Selection of studies

Following searches, the duplicated articles will be removed. Two independent reviewers (RKA and SW) will screen the titles and abstracts of all identified studies. Full-text articles of potentially eligible studies will be retrieved and reviewed independently by two members of the study team (RKA and SW). Any disagreements will be resolved by discussion or, if necessary, by consulting a third review author (NQ/JK) to reach consensus. Studies that fulfil the pre-defined criteria will be included.

## Data extraction and management

Data extraction will be conducted independently by two members of the study team using a standardised and piloted data extraction form for all included studies. The domains for the data extraction form, online supplementary file—appendix 2, are informed by the CHARMS checklist.[12] Each data element will be compared between the primary and secondary reviewers, and any discrepancies will be resolved by discussion, or by adjudication by a third reviewer.

## Risk of bias assessment

Two members of the team will independently assess the risk of bias of the included studies using the Prediction model study Risk Of Bias ASsessment Tool (PROBAST).[16] PROBAST assesses both the risk of bias and concerns regarding the applicability of a study that evaluates a multivariable diagnostic or prognostic prediction model. All four domains (ie, participants, predictors, outcome and analysis) of PROBAST will be used to assess the risk of bias. Any discrepancies will be resolved by discussion or by adjudication by a third reviewer.

## Evidence synthesis

A narrative synthesis approach will initially be used to systematically describe the characteristics and quantitative data from the included studies. Study follow-up periods for the primary outcome(s) of ≤1 year will be categorised as 'short', 1–5 years as 'medium', and above 5 years as 'long-term'.

## Meta-analysis

In articles examining the performance of the same prediction model on various outcomes or multiple time points, we will pool rescaled measures of the predictive performance of the models with similar outcomes using a random-effects meta-analysis using restricted maximum likelihood estimation and applying the Hartung-Knapp-Siddik-Jonkman confidence intervals derivation. Ninety-five per cent (95%) prediction intervals will also be estimated, where possible. Predictive performance of the model will be based on discrimination (such as the c-statistic for binary outcome models, D-statistics for survival outcome models or area under the curve, $R^2$ statistic, Brier score, sensitivity and

specificity or positive and negative predictive values), calibration (total observed events : expected events (O:E) ratio, goodness of fit statistics (such as the Hosmer-Lemeshow goodness of fit test)) and risk reclassification. C-statistics >0.75 and total O:E ratios between 0.8 and 1.2 will be deemed to be of good performance.[11] Additionally, where possible, we will perform multivariate meta-analysis models to jointly synthesis measures of discrimination and calibration. Heterogeneity between studies will be estimated using the $I^2$ statistic for univariate meta-analysis models.

Sensitivity analysis will be done to assess the robustness of the results by excluding studies with a high or unclear risk of bias. We aim to carry out sub-group analyses to explore heterogeneity between studies. If possible, the sub-group analysis will be based on:

1. Index CVD type — coronary heart disease, stroke and PAD.
2. Risk factors — modifiable and non-modifiable factors.
3. Outcomes — primary outcomes (morbidity, mortality).
4. Follow-up duration.
5. Region — based on the Organisation for Economic Co-operation and Development classification—that is, low/middle-income and high-income countries.

*p*-values of 0.05 or lower will be considered to be statistically significant.

## Patient and public involvement

Patients and the public were not involved in the design and conception of this study.

## Ethics and dissemination

Ethical approval and patient informed consent are not necessary because all data will be obtained from previously published studies. We aim to publish our results in a general medical or cardiology peer-reviewed journal to ensure the findings reach a wide readership. We also plan on presenting findings at relevant international conferences.

## DISCUSSION

There have been numerous reviews focussing on primary prevention of CVD.[17 18] To the best of our knowledge, this will be the first systematic review to evaluate existing evidence regarding prognostic models aimed at stratifying CVD severity for secondary prevention. The findings of this review will contribute to the existing literature by identifying the current and most effective prognostic model(s), based on measures of predictive accuracy such as c-statistics,[10] to stratify CVD severity. This will be a significant step towards informing the clinical management of patients with established CVD diagnosis.

This review will also provide an evidence base for the development and validation of future prognostic model(s) to stratify CVD risk severity in patients with an established CVD diagnosis. Prognostic factors found to have important and consistent prognostic value will be included in a related study that aims to develop and validate a risk stratification

model for CVD severity in patients with established CVD diagnosis.

With the significant increase in the number of patients surviving their initial CVD events, a pragmatic means of identifying patients with severe CVD is becoming increasingly important to guide preventive and therapeutic strategies for CVD in the current era of personalised medicine.

**Author affiliations**
[1]Division of Primary Care, University of Nottingham, Nottingham, UK
[2]Division of Epidemiology and Public Health, University of Nottingham, Nottingham, UK
[3]Department of Cardiology, Division Heart & Lungs, University Medical Centre Utrecht, Utrecht University, Utrecht, The Netherlands
[4]Institute of Cardiovascular Science, Faculty of Population Health Sciences, University College London, London, UK
[5]Cardiovascular Research Group, Faculty of Biology, Medicine and Health, University of Manchester, Manchester, UK
[6]Guy's and St. Thomas' NHS Foundation Trust, London, UK

**Acknowledgements** We thank Nia Roberts, Information Specialist with the University of Oxford, for her tremendous support and guidance in developing the search strategies for the various databases.

**Contributors** RKA, NQ, JK and SFW were involved in the study conception. RKA, JL-B, FWA, RSP, PD, ASW, OHI, JK, NQ and SFW have been involved in the design. The protocol was drafted by RKA. All authors (RKA, JL-B, FWA, RSP, PD, ASW, OHI, JK, NQ and SFW) reviewed and approved the final manuscript. RKA is the guarantor of the protocol.

**Funding** RKA is funded by a National Institute for Health Research School for Primary Care Research (NIHR SPCR) PhD Studentship award.

**Competing interests** NQ is a member of the most recent NICE Familial Hypercholesterolaemia and Lipid Modification Guideline Development Groups (CG71 and CG181). SFW is a member of the Clinical Practice Research Datalink (CPRD) Independent Scientific Advisory Committee (ISAC). RKA currently holds an NIHR-SPCR funded studentship (2018–2021). SFW previously held an NIHR-SCPR career launching fellowship award (2015–2018).

**Patient and public involvement** Patients and/or the public were not involved in the design, conduct, reporting, or dissemination plans of this research.

**Patient consent for publication** Not required.

**Provenance and peer review** Not commissioned; externally peer reviewed.

**ORCID iDs**
Ralph K Akyea http://orcid.org/0000-0003-4529-8237
Nadeem Qureshi http://orcid.org/0000-0003-4909-0644

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
