## [Reviewer comments · BMJ Open]

ARTICLE DETAILS

TITLE (PROVISIONAL)	Predicting major adverse cardiovascular events for secondary prevention: Protocol for a systematic review and meta-analysis of risk prediction models
AUTHORS	Akyea, Ralph; Leonardi-Bee, Jo; Asselbergs, Folkert; Patel, Riyaz; Durrington, Paul; Wierzbicki, Anthony; Ibiwoye, Oluwaseun; Kai, Joe; Qureshi, Nadeem; Weng, Stephen

VERSION 1 - REVIEW

REVIEWER	David Kent and Ben Wessler Tufts Medical Center, USA
REVIEW RETURNED	05-Dec-2019

GENERAL COMMENTS	Akyea et al. describe the protocol of an ambitious systematic review of prediction models for patients with established CVD. This work is important as there is substantial redundancy in the literature and clinicians and researchers have significant interest in understanding which models perform best for certain clinical decisions. A major strength of this protocol is the focus on assessing the Prediction model study Risk of Bias Assessment Tool (PROBAST) for each prediction model. Nevertheless, we had some major concerns with this protocol, and especially a lack of detail of some key elements. Some particularly concerning aspects include: • The authors do not detail a search strategy. Since there are no MESH terms that specifically identify clinical prediction models, this is an enormous undertaking.• The authors indicate that they will include patients with established CVD, but the index conditions they mention include only CAD, PAD and Stroke. What about other forms of CVD, including Cardiac surgery sudden death, CHF and arrhythmia. This needs to be specified.• Somewhat less concerning is the lack of specificity to the outcomes. There are numerous cardiovascular outcomes that are missing from the proposed list. Examples include arrhythmias, bleeding, thromboembolic outcomes, heart failure events. Will all outcomes be included or only some?• The authors imply but don't explicitly state whether they will search and include articles that are validation studies, or only articles that derive an original model. To find validation studies search strategies should include citation searches of each of the original clinical prediction models. This alone is a major undertaking.• There should be rigorous attention to inclusion and
--

	exclusion criteria for this review. In particular, the authors need a good definition for a clinical prediction model. It is common that multivariable analyses are presented in a way that is difficult to distinguish whether the intent is to develop a clinical prediction model. What about novel methods of prediction such as machine learning? Additional Concerns: The authors claim that this is the first such review of risk models for patients with established CVD. There is a major effort currently underway that the authors should reference as the proposed review overlaps substantially with our ongoing work (PMID: 31093549, http://pacecpmregistry.org/). We note that in the above work, we have identified >1200 clinical prediction models where either the index condition or the outcome was CVD—and this is only through 2015 (update underway). Unlike the proposed work, we excluded models that were not fully presented in a useable form (such as those presenting regression equations omitting the intercept or baseline hazard). While most models have not been validated, we also documented >1700 external validations thus far. We point this out to emphasize the size of the undertaking. Regarding PROBAST, the authors should indicate how they will establish inter-reader agreement in this new tool. In our experience, the full tool takes ~40 to 45 minutes to apply to each study and there is substantial inter-reader disagreement on many of the individual items. Additionally, the tool classifies >90% of studies as “high risk of bias” since the threshold for high risk is very low. Modification of the tool may be needed for this ambitious project. In the context of this study, it is unclear what the investigators mean by “second level of evidence from observational cohort studies” in their limitations section. It is not clear how the authors will use their findings to achieve their aim of “identifying... the most effect prognostic models”
--	---

REVIEWER	George Siontis Department of Cardiology, University Hospital of Bern, Switzerland.
REVIEW RETURNED	08-Dec-2019

GENERAL COMMENTS	 - Abstract-Methods: Should be made clear that eligible studies, are studies in which a prediction models is presented (derived/validated). - Abstract-Methods: "Eligible studies are prospective cohort studies... ". Why do the authors focus only on prospective studies? Studies dealing with prediction models in retrospective cohorts should be also included. The same applies to Methods Section. - Selection criteria-Patient group: “with an established diagnosis of CVD”: the authors should provide further details how they define “CVD”. - The prespecified data extraction items should be reported. - The authors should clarify how they will define that the same prediction model has been used across different studies. Based on which criteria? It is important to clearly define this, since they are planning to subsequently meta-analyse those models presented in different validation studies.
---

VERSION 1 – AUTHOR RESPONSE

Response to Comments from Reviewer 1

Specific comments

The authors do not detail a search strategy. Since there are no MESH terms that specifically identify clinical prediction models, this is an enormous undertaking.

Author response: Appendix 1 (now moved to Supplementary File) of our submitted manuscript detailed our full search strategy with specific MESH terms for all the domains outlined in the Methods section: cardiovascular disease, risk scores, and predictive performance assessment. These search strategies were adapted from published reviews and developed in consultation with an Information Specialist from the Oxford Centre for Evidence-based Medicine (<https://www.cebm.net/>).

The authors indicate that they will include patients with established CVD, but the index conditions they mention include only CAD, PAD and Stroke. What about other forms of CVD including Cardiac surgery sudden death, CHF and arrhythmia. This needs to be specified.

Author response: Thanks for the suggestion. However, as indicated in our discussion, the review will provide the evidence base for the development and validation of future prognostic models to stratify risk in patients with established cardiovascular disease (CVD) diagnosis. The study population has been specifically focussed on the major CVD events (that is, coronary artery disease, cerebrovascular artery disease and peripheral artery disease [1,2]) in consultation with the team of collaborating Consultant Cardiologists who are experts in the field of review. Also, the reviewers acknowledge below the vast number of papers (> 1000), hence why a review in this field of research should and must be highly focussed.

Somewhat less concerning is the lack of specificity to the outcomes. There are numerous cardiovascular outcomes that are missing from the proposed list. Examples include arrhythmias, bleeding, thromboembolic outcomes, heart failure events. Will all outcomes be included or only some?

Author response: Thanks once again for the suggestion. Our review is not aimed at looking at all possible outcomes related to CVD. Major adverse cardiovascular events (MACE) as defined in the detailed primary outcome section are the main outcomes of interest [3]. This is informed by our aim to generate evidence for our subsequent projects. We defined MACE by the most common CVD outcomes and used as primary outcomes in most clinical studies (TIMI trials, <https://www.nejm.org/doi/full/10.1056/NEJMoa1812389> and CTT trials, <https://www.ahajournals.org/doi/10.1161/JAHA.118.009221>). Again, this is to focus the review. Other outcomes mentioned, whilst important, are what we believe should be separate reviews and beyond the scope of the main outcomes of interest in this study. The section on primary and secondary outcomes (page 6) have been revised to provide more clarity.

The authors imply but don't explicitly state whether they will search and include articles that are validation studies, or only articles that derive an original model. To find validation studies search strategies should include citation searches of each of the original clinical prediction models. This alone is a major undertaking.

Author response: As rightly indicated by the reviewers, the review will consider articles reporting both original models and validation studies. This is explicitly stated under the study design section (page 5):

- i. Published as an original research article (that developed, compared or validated a multivariable prognostic model or clinical prediction rule) in a peer-reviewed journal.

The aim of the review is to first, identify all prognostic models that have been published. It is, therefore, impractical to pre-specify original models as we might miss out on models we do not know about, hence introducing a bias in our review. To avoid this, we used MESH terms for predictive performance assessment,[4,5] as both original and validation studies will report one of these assessment measures.

There should be rigorous attention to inclusion and exclusion criteria for this review. In particular, the authors need a good definition for a clinical prediction model. It is common that multivariable analyses are presented in a way that is difficult to distinguish whether the intent is to develop a clinical prediction model. What about novel methods of prediction such as machine learning?

Author response: We agree with the reviewers, for some published articles there might be some uncertainty in what the authors intent was. To help address this issue amongst others, the screening and assessment are being done by two independent researchers, with a third experienced and senior researcher resolving any disagreements through a team discussion. To clarify and be more explicit about our definition of what constitutes a risk prediction model, which is multivariable analyses, we have included the word multivariable (page 5) under selection criteria section.

Additional Concerns: The authors claim that this is the first such review of risk models for patients with established CVD. There is a major effort currently underway that the authors should reference as the proposed review overlaps substantially with our ongoing work (PMID: 31093549, <http://pacecpmregistry.org/>). We note that in the above work, we have identified >1200 clinical prediction models where either the index condition or the outcome was CVD—and this is only through 2015 (update underway). Unlike the proposed work, we excluded models that were not fully presented in a useable form (such as those presenting regression equations omitting the intercept or baseline hazard). While most models have not been validated, we also documented >1700 external validations thus far. We point this out to emphasize the size of the undertaking.

Author response: We are grateful to the reviewers for sharing their previous work with us and sharing their experience with us. We will want to highlight that although there might be some similarities or minor overlaps, we are addressing separate research questions. The Reviewers suggest > 1200 clinical prediction models but is including a broader inclusion criterion “where the index condition or outcome was CVD”. This has been well-established that there are a large number of studies primary prevention studies published previously in the BMJ (<https://www.bmj.com/content/353/bmj.i2416>) identifying 363 prediction models and 473 external validations. Our research question is highly focussed on only secondary prevention models. A possible overlap does not invalidate either our study or the update of the previous work of the reviewers as there are clear distinctions in the work being undertaken.

Regarding PROBAST, the authors should indicate how they will establish inter-reader agreement in this new tool. In our experience, the full tool takes ~40 to 45 minutes to apply to each study and there is substantial inter-reader disagreement on many of the individual items. Additionally, the tool

classifies >90% of studies as “high risk of bias” since the threshold for high risk is very low. Modification of the tool may be needed for this ambitious project.

Author response: To establish an inter-reader agreement, two reviewers will apply PROBAST tool to each included study/article with a third reviewer resolving any disagreements as per standard systematic review methods.

PROBAST is a recent risk of bias tool which is peer-reviewed, has been recommended and used in a number of published articles in high impact journals such as the BMJ recently for similar prognostic model reviews in different disease conditions[5,6]. Like every other tool available, it has its limitations. We do not think it is within our capability to modify and validate a modified PROBAST tool within our current study. As with all tools, we acknowledge that there may be some limitations of the tool itself - it will be most expedient for the reviewers to consider developing a modified PROBAST tool to address the issues they have explicitly outlined for peer-review.

In the context of this study, it is unclear what the investigators mean by “second level of evidence from observational cohort studies” in their limitations section.

Author response: Thanks for highlighting this. The sentence has been revised to provide clarity. It now reads (page 3): “Evidence from observational cohort studies may be used in this context and this level of evidence may, therefore, be subject to bias and confounding.”

It is not clear how the authors will use their findings to achieve their aim of “identifying... the most effect prognostic models”

Author response: Measures of predictive accuracy such as c-statistics and calibration parameters can be used to assess and compare model performance. (doi: <https://doi.org/10.1136/bmj.e3318>)

Response to Comments from Reviewer 2

Specific comments

Abstract-Methods: Should be made clear that eligible studies, are studies in which a prediction models are presented (derived/validated).

Author response: Thanks for the suggestion. We have revised the Abstract – Methods. Now reads: “Eligible studies present multivariable models (derived or validated) of adults (aged 16 years and over) with an established diagnosis of CVD, reporting at least one of the components of the primary outcome of major adverse cardiovascular events (defined as either coronary heart disease, stroke, peripheral artery disease, heart failure or CVD-related mortality).

Abstract-Methods: "Eligible studies are prospective cohort studies... ". Why do the authors focus only on prospective studies? Studies dealing with prediction models in retrospective cohorts should be also included. The same applies to Methods Section.

Author response: We agree this needs to be clarified. This has been revised accordingly.

Selection criteria-Patient group: “with an established diagnosis of CVD”: the authors should provide further details how they define “CVD”.

Author response: Thanks for highlighting this. We have revised the sentence to define CVD and provide better clarity. The sentence now reads (page 5): “Adults, 16 years and above, with an established diagnosis of CVD (where CVD is defined as a documented clinical diagnosis of arterial occlusive events including coronary artery disease, cerebrovascular artery disease and peripheral artery disease (PAD)).[8,9]”

The prespecified data extraction items should be reported.

Author response: Thanks for the recommendation. We have included the prespecified data extraction form as Appendix 2 in the Supplementary File.

The authors should clarify how they will define that the same prediction model has been used across different studies. Based on which criteria? It is important to clearly define this, since they are planning to subsequently meta-analyse those models presented in different validation studies.

Author response: We agree with the reviewer it is important to clearly define this. The text has been revised (page 7): “In articles examining the performance of the same prediction model on various outcomes or multiple timepoints, we will pool rescaled measures of the predictive performance of the models with similar outcomes using a random-effects meta-analysis using restricted maximum likelihood (REML) estimation and applying the Hartung-Knapp-Siddik-Jonkman confidence intervals derivation.”

VERSION 2 – REVIEW

REVIEWER	George Siontis Department of Cardiology, University Hospital of Bern, Switzerland.
REVIEW RETURNED	27-Jan-2020

GENERAL COMMENTS	All my comments have been adequately addressed.
---